# Comparative Effectiveness of Non-Pharmacological and Pharmacological Treatments for Non-Acute Lumbar Disc Herniation: A Multicenter, Pragmatic, Randomized Controlled, Parallel-Grouped Pilot Study

**DOI:** 10.3390/jcm14041204

**Published:** 2025-02-12

**Authors:** Doori Kim, Jee Young Lee, Yoon Jae Lee, Chang Sop Yang, Chang-Hyun Han, In-Hyuk Ha

**Affiliations:** 1Jaseng Hospital of Korean Medicine, 536 Gangnam-daero, Gangnam-gu, Seoul 06110, Republic of Korea; doori.k07@gmail.com; 2Department of Korean Internal Medicine, Integrative Cancer Center, Cha Ilsan Medical Center, 1205, Jungang-ro, Ilsandong-gu, Goyang-si 10414, Republic of Korea; happiade@hanmail.net; 3Jaseng Spine and Joint Research Institute, Jaseng Medical Foundation, 540 Gangnam-daero, Gangnam-gu, Seoul 06110, Republic of Korea; goodsmile8119@gmail.com; 4KM Science Research Division, Korea Institute of Oriental Medicine, 1672, Yuseong-daero, Yuseong-gu, Daejeon 34054, Republic of Korea; yangunja@kiom.re.kr; 5School of Korea Institute of Oriental Medicine, Korean Convergence Medical Science, University of Science & Technology, Daejeon 34054, Republic of Korea

**Keywords:** back pain, sciatica, pragmatic randomized study, pharmacological therapy, manual therapy, spinal manipulation

## Abstract

**Background/Objectives:** We aimed to compare non-pharmacological (non-PHM) and pharmacological (PHM) treatment for patients with non-acute lumbar disc herniation (LDH) and determine the feasibility of a large-scale study. **Methods:** This was a two-armed, parallel, multicenter, pragmatic controlled trial performed in South Korea. All patients underwent magnetic resonance imaging (MRI) scans both at the screening stage and the last follow-up. Patients with LDH findings on MRI were randomly assigned to non-PHM and PHM groups. Treatment was administered twice a week for a total of 8 weeks, and follow-up assessments were performed at weeks 9, 13, and 27 post-randomization. The primary outcome was the Oswestry Disability Index (ODI) score. A linear mixed model was used for primary analysis from intention-to-treat perspectives. The incremental cost-effectiveness ratio (ICER) was calculated for economic evaluation. **Results:** Thirty-six patients were enrolled, and thirty-five were included in the final analysis. At Week 9, the difference in ODI scores between the two groups was 5.17 (95% CI: −4.00 to 14.35, *p* = 0.262), and the numeric rating scale scores for lower back and leg pains were 1.89 (95% CI: 0.68 to 3.10, *p* = 0.003) and 1.52 (95% CI: 0.27 to 2.77, *p* = 0.018), respectively, confirming greater improvement in the non-PHM group than in the PHM group. The non-PHM group showed lower costs and higher quality-adjusted life years than the PHM group. The ICER calculated using the EuroQoL-5 Dimension (EQ-5D) was USD 20,926. **Conclusions:** We confirm the possibility that a non-PHM strategy could be a more effective and cost-effective treatment option than PHM for patients with non-acute lumbar disc herniation. Furthermore, this pilot study confirmed the feasibility of the main study in terms of design and patient compliance.

## 1. Introduction

Low back pain (LBP) is one of the most common symptoms of musculoskeletal conditions, with a lifetime prevalence as high as 85% [1,2], and is a leading cause of years lived with disability worldwide [3,4]. Sciatica is characterized by pain radiating along the sciatic nerve, running down the leg from the lower back, and a sensation of numbness or tingling [5]. The prevalence of sciatica is variable and not as high as that of LBP [6]; however, it can still affect up to 40% of the population during their lifetime in terms of their physical and mental health and can cause a significant economic burden not only at the level of individual patients but also across society [7,8,9,10]. Lumbar disc herniation (LDH) is one of the most common causes of low back pain and sciatica [11]. Sciatica from acute LDH shows a favorable natural course, with recovery taking from 2 weeks to 3 months in most cases [12,13]; however, for some patients, pain persists for more than 12 months [13], and for others, it turns into a chronic condition and a lifelong experience with recurrent symptoms [14].

Conservative management is the first-line therapy for sciatica, but no rule has been agreed upon regarding the methods of conservative treatment, and guidelines show mixed views in their recommendations [13]. The American College of Physicians (ACP) recommends that clinicians and patients should initially select non-pharmacological (non-PHM) treatments such as superficial heat therapy, massage, acupuncture, exercise therapy, or spinal manipulation for patients with acute or chronic LBP with or without sciatica [15]. The Korean Medicine Clinical Practice Guidelines for Lumbar Herniated Intervertebral Disks in Adults [16] also present acupuncture and spinal manipulation with high-grade recommendations. However, guidelines from the United Kingdom [17] and Denmark [18] recommend against acupuncture and do not strongly recommend spinal manipulation. These existing guidelines also present different grades of recommendation for invasive procedures, such as epidural injections or lumbar facet joint blocks [15,17,18,19].

Major guidelines recommend the prudent use of PHM treatment [20]. The ACP [15] recommends that PHM treatment should be considered only when patients have an inadequate response to first-line non-PHM therapies. The United Kingdom guidelines also recommend that to minimize harm, medications should be considered with minimal doses over a short course [17]. Opioids should only be considered in carefully selected patients and should be used in short courses [21,22].

However, there are gaps between real-world clinical practice and recommendations in these guidelines. The use of opioids for LBP is common in the United States and Canada, and the number of opioid prescriptions has been increasing [23,24]. In a retrospective analysis conducted in South Korea, 78.5% of patients with intervertebral disc disorder (IDD) were prescribed non-opioid analgesics, and more than 14% of patients with IDD underwent epidural nerve blocks [25]. Injection therapy is used in 10–11% of patients with LBP [26]. These reports indicate that in real-world clinical settings, active PHM therapy, including nerve blocks, is frequently used, unlike the recommendations in clinical practice guidelines.

Amid conflicting views among different guidelines and discrepancies between the recommendations and treatment methods applied in real-world clinical practice, there is a pressing need for high-level evidence to support the selection of optimal treatment strategies. South Korea has a dual system of medical licenses for conventional (Western) and Korean medicine (KM). KM doctors can perform acupuncture and conservative manual therapy, such as spinal manipulation therapy, as an equivalent or alternative treatment modality to injection therapy in Western medicine. Patients can select their treatment methods based on their preferences and conditions. Therefore, it can be argued that South Korea has a system and environment of medical services in which non-PHM treatment strategies, such as acupuncture and manual therapy, can be compared with PHM treatment strategies in real-world clinical settings.

Therefore, we aimed to compare the clinical effectiveness, cost-effectiveness, and safety of a non-PHM treatment strategy that includes acupuncture, electroacupuncture, and spinal manipulation therapy with a PHM treatment strategy for active medical management that includes medications and injections, reflecting real-life clinical practice. The research team previously conducted a small-scale pilot study with a sample size of 30 patients at a single institution [27]. However, in this previous study, both groups received treatment at the same study institution, which was a KM hospital, and the study design of the participants receiving PHM treatment at a KM hospital did not properly reflect real-world clinical practice.

Therefore, it was decided not to place restrictions on treatment institutions and allow patients to freely receive PHM treatment at the institution of their choice. In addition, to facilitate the process of patient recruitment required for a large-scale trial, four hospitals participated as study institutions rather than as single centers as in the previous study. Since new study institutions were added and the research team was not familiar with the new study process of allowing treatment from external institutions other than the study institutions, the necessity for additional evaluation and confirmation of feasibility was suggested. Accordingly, we performed a comparative evaluation between non-PHM and PHM treatment strategies for patients with non-acute LDH with LBP or sciatica (radiating leg pain) and, as a pilot study, aimed to evaluate the feasibility of conducting the subsequent main trial.

## 2. Materials and Methods

### 2.1. Study Design and Setting

This was a multicenter, pragmatic, randomized, controlled, parallel-grouped pilot trial. From four KM hospitals in South Korea, 36 patients (9 from each hospital) were recruited through noncompetitive recruitment. The participants were randomly assigned at a ratio of 1:2 to the non-PHM treatment group (*n* = 12) and the PHM treatment group (PHM group) (*n* = 24). The participants underwent treatment sessions twice a week for a total of 8 weeks, and follow-up assessments were performed at weeks 9, 13, and 27 post-randomization (Figure 1).

Before the initiation of the trial, the researchers (investigators) held one-on-one meetings with individual participants, provided a full explanation of the clinical study (information related to effectiveness, adverse events, and safety), and obtained signed consent forms from the participants. The study was conducted following the Declaration of Helsinki. All documentation related to the study, including the study protocol, was approved by the Institutional Review Board of the Jaseng Hospital of Korean Medicine (JASENG 2021-05-016, JASENG 2021-05-018, JASENG 2021-05-019, and JASENG 2021-05-023), and date of approval was 6 June 2021. In addition, the protocol of this study was published [28], and the trial was registered at Clinicaltrials.gov (NCT05003726, registration date 12 August 2021).

### 2.2. Participants

The inclusion criteria were as follows: (1) age 19–69 years; (2) radiating leg pain (sciatica) that first developed not less than 3 months before enrollment; (3) numeric rating scale (NRS) score of radiating leg pain or LBP of ≥5 for 3 consecutive days; (4) radiologic diagnosis of LDH on magnetic resonance imaging (MRI), which can explain the radiating leg pain (sciatica) and LBP; and (5) those who gave their consent to study participation and signed an informed consent form. The exclusion criteria are presented in Appendix A.

### 2.3. Intervention

Patients assigned to each group were first educated about non-PHM and PHM treatment strategies and then received the applicable treatments. Next, according to the respective treatment strategies, patients received treatment based on the medical judgment of clinicians from the study and external medical institutions. Patients in each group were recommended to receive the assigned treatment, but there were no specific restrictions on receiving additional treatments other than the recommended treatment strategy.

### 2.4. Non-PHM Treatment Group

The basic treatment schedule for KM non-PHM treatment was a total of 16 treatment sessions consisting of two sessions per week for 8 weeks; however, the number of treatments for each participant was adjusted according to the condition of individual patients and the medical judgment of the clinicians. The representative modalities of non-PHM treatment used in this study were acupuncture, electroacupuncture, and spinal manual therapy [29]. The acupoints used in acupuncture therapy, depth of needle insertion, frequency and intensity of electroacupuncture, and type of manual therapy applied to individual patients were selected according to the clinical judgment of the KM doctors in charge of the treatment. The details of treatment were recorded in the case report forms.

### 2.5. PHM Treatment Group

Patients assigned to the active comparator group (PHM treatment strategy group) received treatment without restrictions by selecting departments, such as neurosurgery or orthopedics, of external clinics or institutions. Methods of PHM treatment included the prescription of medications and nerve blocks. The basic guidelines were to receive nerve blocks three or more times during the total intervention period of 8 weeks, but the details of the actual procedure were determined according to the medical judgment of the clinicians. The basic guidelines for medications were to visit the selected clinic/hospital twice a week for a prescription; however, the actual details of the number of visits, prescription status, and days of prescription filling were determined according to the medical judgment of the clinicians. Depending on the condition of each patient, the administration of physiotherapy, such as transcutaneous electrical nerve stimulation or interferential current therapy (ICT), was allowed. After each treatment visit, patients were issued a receipt with treatment details, and by referring to the receipts issued to and brought by the patients, the method of procedure used, type of prescribed medications, number of prescription days, dose, and route of administration were recorded in the case report form.

### 2.6. Outcomes

#### 2.6.1. Primary and Secondary Outcomes

The primary outcome of this study was the change in the Oswestry Disability Index (ODI) score from baseline to the primary endpoint, which was week 9. A validated Korean version of the ODI was used [30]. ODI scores were measured five times in total: at baseline, week 5, and three time points during the follow-up period (weeks 9, 14, and 27) (Appendix A).

The secondary outcomes were the NRS and visual analog scale (VAS) scores for LBP and radiating leg pain, patient global impression of change (PGIC), Short Form-12 Health Survey version 2 (SF-12 v2), and 5-Level EuroQoL-5 Dimension (EQ-5D-5L). Further details on each outcome and time point of assessment are presented in Appendix A.

#### 2.6.2. Economic Evaluation

For the collection and analysis of costs to perform the economic evaluation, a structured questionnaire was developed to collect information on formal/informal medical costs, non-medical costs, time costs, and productivity loss costs. Formal medical costs are those incurred from using medical services in medical institutions, and informal medical costs are those spent on informal healthcare, such as purchasing functional health foods and medical devices. Non-medical costs include transportation costs, patient time costs, and costs for caregiver services, which are secondary to the use of medical services. Productivity loss costs refer to the amount of economic loss incurred by not being able to engage in a job because of the disease itself or premature death caused by the disease.

Medical costs were calculated using patient surveys, data from the Health Insurance Review and Assessment Service, and data from medical institutions. For the estimation of quality-adjusted life year (QALY) scores, EQ-5D-5L and SF-6D were used, and QALYs were calculated using the area under the curve method. The productivity loss costs were calculated using a Work Productivity and Activity Impairment questionnaire (WPAI) [31].

Information about medical and time costs was collected for all visits, and time costs were collected once in week 2, based on the assumption that the same length of time was spent on each visit. Information on productivity loss was collected at baseline, once a week during the intervention period, and thrice during the follow-up period (Appendix A).

#### 2.6.3. Drug Consumption

The types and doses of medications (prescribed for the present illness or rescue medications) taken during the study period were checked through a patient survey at each visit. All prescribed medications, as well as over-the-counter medications, were investigated for both the PHM and non-PHM groups. Other treatments, such as physiotherapy and injection therapy, apart from medications taken, were recorded using the frequency of the treatments received. Information on drug consumption was collected during each visit.

### 2.7. Sample Size Estimation

In this pilot study, to evaluate the feasibility of the main follow-up study, the randomization ratio was set to 1:2 to obtain as much control group information as possible. Based on a previous report that a minimum sample size of 12 was needed for a pilot study, the minimum unit was set to 12 [32]; therefore, the target sample size for recruitment was 36 (12 participants in the non-PHM treatment strategy group (non-PHM group) vs. 24 participants in the PHM group).

### 2.8. Randomization and Allocation Concealment

Participants who fulfilled the inclusion and exclusion criteria of the screening process and provided their consent to participate by signing the informed consent form were allocated to either of the two groups in a 1:2 ratio (12 participants vs. 24 participants) using a randomization table. The table was generated by a statistician using R Studio 1.1.463 (2009–2018 RStudio, Inc., Boston, MA, USA). Permuted block randomization was performed to generate random sequences. The size of a single block was set to three, and patients were allocated to either the experimental or control group in a 1:2 ratio. The nine study participants were stratified and allocated to one study institution. The randomization results were sealed in an opaque envelope and stored in a cabinet with double locks. The investigator at each study site opened the randomization envelope and allocated the participants into groups. The randomization number assigned to each participant was recorded in an electronic chart.

### 2.9. Blinding

Owing to the study design, blinding was not possible, and this was an open-label study. Assessor blinding was implemented in this setting. The assessors did not participate in the intervention but performed assessments in a separate place while blinded.

### 2.10. Statistical Analysis

In this study, an intention-to-treat (ITT) analysis was the primary analysis, and a per-protocol (PP) analysis was performed by including participants who visited medical institutions six or more times during the intervention period. To handle missing values, in the case of the linear mixed-model approach, a method for primary analysis, the Mixed Model for Repeated Measures, was used for data processing. For sensitivity analysis, the last observation carried forward (LOCF) and multiple imputation (MI) were used for missing data processing.

The sociodemographic characteristics and treatment expectations of the participants were evaluated for each group. Continuous variables are expressed as mean (and standard deviation [SD]) or median (and quartiles), and differences between the two groups were tested using an independent *t*-test or Wilcoxon rank sum test. Categorical variables were expressed as frequency (and percentage [%]), and differences between the two groups were compared using the chi-squared test or Fisher’s exact test.

The effectiveness endpoint was the change in continuous outcomes (NRS, VAS, ODI, EQ-5D-5L, and SF-12 scores) from baseline to each time point. For the primary analysis, a linear mixed model that considers the baseline value of each variable and variables considered to show clinically significant changes from the baseline among covariant factors with significant between-group differences as covariates and a group as a fixed factor was used. For sensitivity analysis, analysis of covariance (ANCOVA) was performed for the sets processed using MI and LOCF.

To compare the cumulative values of the difference in each outcome within the period (intervention and total study periods) in the two groups, the areas under the curve at each time point post-randomization were calculated. If the superiority test failed, a non-inferiority test was performed. The non-inferiority margin was set to −5.5, which is half of the ODI minimal clinically important difference for LDH. If the lower bound of the 95% confidence interval (CI) for the difference in the reduction of ODI between the two groups was not greater than the margin, non-PHM treatment was considered not inferior to PHM treatment. In all analyses, the significance level was set at 0.05, and SAS 9.4 (SAS Institute, Inc., Cary, NC, USA) or R Studio 1.1.463 (2009–2018 RStudio, Inc.) was used for all statistical analyses.

In addition to comparing the clinical effectiveness and safety, an economic evaluation was performed to compare the cost-effectiveness of non-PHM treatment and PHM treatment as active comparators. The primary measure of economic evaluation of the ICER of the non-PHM treatment group to that of the PHM treatment group was used. The ICER was assessed by calculating the ratio of the costs per QALY. The uncertainty of cost-effectiveness was estimated using bootstrap sampling for cost and QALYs. In addition, the probability that the non-PHM treatment strategy was cost-effective according to willingness to pay (WTP) was tested using cost-effectiveness acceptability curves. All economic evaluation analyses were performed using costs calculated from the healthcare system and societal perspectives.

### 2.11. Adverse Events (AEs)

All AEs occurring during the trial were recorded, referring to all untoward or unintended signs (e.g., abnormalities in laboratory test results), symptoms, or diseases occurring after performing interventions during the trial. The definition included events without a causal relationship with the applied intervention. AEs were recorded through patient complaints and investigator observations. For all AEs, the causal relationship between the treatment and AEs was assessed using a six-level scale: 1, definitely related; 2, probably related; 3, possibly related; 4, probably unrelated; 5, definitely unrelated; and 6, unknown. The severity of AEs was assessed at three levels according to the Spilker classification as follows: mild (1), did not impair the participants’ normal activities of daily living (ADLs), caused minimal discomfort, and required no additional treatment; moderate (2), significantly impaired the participants’ normal ADLs and may have required treatment but was resolved after treatment; and severe (3), severely impaired the participants’ normal ADLs, required intense treatment, and left sequelae.

## 3. Results

### 3.1. Participants

Between October 2021 and November 2022, 303 patients were screened at the four KM hospitals in South Korea. Of these, 36 patients who met the inclusion criteria and provided consent to participate in the study were enrolled and randomly assigned to non-PHM and PHM groups at a ratio of 1:2. One participant in the non-PHM group could not be contacted during the intervention period, one participant in the PHM group withdrew consent, and one participant discontinued the intervention because a disease that may affect the pain outcome in this study was identified during treatment. The ITT analysis was conducted on 35 participants (12 in the non-PHM group and 23 in the PHM group), excluding one participant who withdrew their consent (Figure 1).

### 3.2. Baseline Characteristics

The baseline characteristics of the study participants are presented in Table 1. Of the 35 participants, 19 (54.3%) were female, indicating a similar ratio between male and female participants. The mean age was 52.08 ± 10.98 years in the non-PHM group and 44.26 ± 11.05 years in the PHM group, indicating that participants of the non-PHM group were slightly older (*p* = 0.055) than those in the PHM group. In terms of radiological findings, four patients (33.3%) in the non-PHM group and seven patients (30.4%) in the PHM group showed extrusion of the intervertebral discs on MRI, and there was no significant difference between the two groups in the outcomes measured at baseline.

### 3.3. Details of Treatment by Group

The details of the treatments received during the intervention period for each group are presented in Table 2. Patients in the non-PHM group received acupuncture (mean 10.8 ± 4.6 times), electroacupuncture (mean 10.8 ± 4.6 times), and Chuna manual therapy (mean 10.7 ± 4.6 times). As for the PHM group, during the intervention period, nine patients (40.9%) received nerve blocks an average of 3.4 ± 4.0 times, and six patients (27.3%) received injections an average of 2.5 ± 1.2 times. In addition, 18 patients (78.3%) were prescribed oral medications for an average of 26.6 ± 15.7 prescription days. The frequently used acupoints and types of Chuna manual therapy in the non-PHM group are presented in Appendix A, and the frequently used medications in the PHM group are shown in Appendix A.

### 3.4. Primary and Secondary Outcomes

The changes in primary and secondary outcomes analyzed using the linear mixed model are shown in Table 3 and Appendix A. The outcomes were adjusted for the baseline values and age. The ODI score, the primary outcome in this study, did not show a significant difference between the two groups at the primary endpoint, week 9 (difference 5.17, 95% CI: −4.00 to 14.35, *p* = 0.262). The lower bound of the confidence interval for the difference between the two groups was −4.00, which was larger than the predefined non-inferiority margin of −5.5, indicating that KM non-PHM treatment met the non-inferiority criteria.

Analysis of the secondary outcomes reveals that the non-PHM group showed significant improvements in LBP and leg pain scores compared to the PHM group. At week 9, the difference in the NRS scores of LBP between the two groups was 1.89 points (95% CI: 0.68 to 3.10, *p* = 0.003), and the difference in the NRS scores of leg pain was 1.52 points (95% CI: 0.27 to 2.77, *p* = 0.018). The EQ-5D-5L, PCS, and MCS scores were also higher in the non-PHM group; however, the differences between the two groups were not statistically significant. Similar trends were observed for the PP and ANCOVA results (Appendix A).

The results of the area under the curve (AUC) analysis by calculating the cumulative outcomes for the total follow-up period of 27 weeks are shown in Figure 2 and Appendix A. The AUC value of the NRS scores for LBP differed significantly between the two groups (difference −34.35, 95% CI: −62.45 to −6.24, *p* = 0.018).

### 3.5. MRI Changes

All patients who participated in the trial underwent MRI scans at screening and final follow-up. As a result of analyzing the severity of LDH in patients who had MRI results both at baseline and follow-up, the number of patients showing disc extrusion at screening was 3 (30.0%) in the non-PHM group and 5 (27.8%) in the PHM group. At the follow-up time point of week 27, the number of patients showing disc extrusion showed no change in the non-PHM group, and the number was reduced to three (16.7%) in the PHM group (Appendix A).

### 3.6. AEs

The AEs reported for each group are shown in Appendix A. Among the seven cases in the PHM group and two cases in the non-PHM group, there was one case of AE that could be related to the intervention, which was a case of headache in the PHM group; its severity was mild. Examples of AEs that were definitely not related to the intervention included stomatitis and cold. The severity of all AEs was mild or moderate, and there were no serious AEs.

### 3.7. Economic Evaluation

Comparing the costs between the two groups, the non-PHM group spent USD 108 (95% CI: −235 to 412) more on medical costs and USD 171 (95% CI: 74 to 272) more on non-medical costs than the PHM group, and the cost of productivity loss was smaller by USD 961 (95% CI: −3301 to 1703) in the non-PHM group compared to that in the PHM group. The total cost from a societal perspective was USD 7493 (95% CI: 5511–9842) in the non-PHM group and USD 8483 (95% CI: 7420–9671) in the PHM group, indicating that the non-PHM group had costs lower by USD 990 (95% CI: −3338 to 1676). The difference in QALYs between the two groups was 0.005 (95% CI: −0.031 to 0.042) when calculated using EQ-5D score and 0.007 (95% CI: −0.023 to 0.037) when calculated using SF-6D scores, indicating that the non-PHM group had greater QALYs (Table 4).

The results of the economic evaluation from the societal and healthcare system perspectives for the two groups are presented in Table 5. From a societal perspective, the non-PHM treatment strategy was confirmed to be the dominant option, with higher QALY values and lower total costs than the PHM treatment strategies. From the healthcare system perspective, which only included medical costs for cost calculation, the ICER calculated using the EQ-5D was USD 20,926, and the ICER calculated using the SF-6D was USD 15,260.

The probability that a non-PHM treatment strategy is cost-effective based on the WTP was 81.9% from a societal perspective and 50.4% from a healthcare system perspective from calculations using the EQ-5D-5L and 83% from a societal perspective, and 57.7% from a healthcare system perspective from calculations using the SF-6D (Table 5 and Figure 3).

## 4. Discussion

This multicenter pilot study was conducted to compare KM-based non-PHM and active PHM treatment strategies in a real-world clinical practice setting for non-acute LDH patients. The non-PHM group showed significantly superior improvement in outcomes in terms of LBP NRS and leg pain NRS scores compared to the PHM group. In addition, in the results of the economic evaluation from a societal perspective, the non-PHM group showed lower total cost and higher QALYs than the PHM group. Furthermore, from a healthcare system perspective, the non-PHM group showed good ICER values, indicating that the KM-based non-PHM treatment strategy can serve as a cost-effective treatment option.

We conducted a single-center pilot prior to this study. The previous study confirmed the possibility of a non-PHM treatment strategy as a more cost-effective treatment strategy than a PHM treatment strategy [27]. However, in this pilot study, additional considerations were required.

In the previous single-center pilot study [27], patients were recruited at a KM hospital, which was the study institution, and both non-PHM treatment and PHM treatment groups received treatment at the KM hospital. However, receiving PHM treatment from MDs in KM hospitals was determined to be inadequate, as the study design reflected real-world clinical practice. In fact, the PHM group received treatment consisting only of nerve blocks, medications, and physiotherapy (ICT, ultrasound, traction, and laser therapy), which was different from the actual treatments received by patients who mainly take medications for their treatment [25]. In addition, patients recruited from KM hospitals exhibited a preference for non-PHM treatment and showed disappointment when assigned to the PHM treatment group; it was difficult to lead these patients to comply properly with the study procedures.

As these difficulties and areas of improvement were identified, it was deemed necessary to conduct another pilot study. In particular, it was necessary to examine patient compliance, such as whether the patients in the PHM group were properly receiving treatment at external medical institutions according to the treatment strategy to which they were assigned and whether the treatment details at external medical institutions could be recorded and collected without problems. That is, since there was a greater need to verify feasibility in the PHM treatment (control) group than in the non-PHM treatment (experimental) group, the ratio of the non-PHM group to the PHM group was set at 1:2 for this multicenter pilot study.

Looking into the treatment details collected during the intervention period, the PHM treatment group received many different types of treatments, such as nerve blocks, injections, medications, manual therapy, and physiotherapy. In real-world clinical practice, especially in clinical settings in South Korea, it is common for patients who mainly use medications for their treatment to receive non-PHM treatments, such as physiotherapy or manual therapy [25]. Therefore, it is reasonable that the treatment details of the PHM treatment group in this pilot study reflected real-world clinical practice better than those in the first pilot study [27].

The ODI score was used as the primary outcome in this study, but there was no significant difference between the two groups at week 9 at the end of the intervention period, thus failing to demonstrate the superiority of non-PHM treatment. However, the lower bound of the CI of the difference between the two groups was −4.00, which was larger than the predefined non-inferiority margin of −5.5, indicating that the non-PHM treatment meets the non-inferiority criteria. The LBP NRS scores, one of the key outcomes other than the ODI scores, were in the range of 2 points in the non-PHM group after interventions and as high as 4 points in the PHM group, showing a significant difference between the two groups. A significant difference was also observed in the leg pain NRS scores. In a previous single-center pilot study, the outcome measured in ODI scores showed some differences between the two groups, but the NRS scores were similar between the two groups. The difference in the results of the two clinical studies with similar designs may have been due to small sample sizes.

Another point of consideration is that in the previous single-center pilot [27], the pain outcomes of the PHM group showed a rapid increase after the completion of all interventions, and the pain was reduced again after some time. For this observation, it was necessary to examine whether this phenomenon was specific to that study or whether it occurred in cases of PHM treatment strategies. In this pilot study, although the degree of change was smaller than that in the first pilot study, the NRS scores also increased slightly after the completion of all interventions and then decreased in the PHM group. Therefore, the above phenomenon may have occurred because the pain once suppressed by medications during the intervention period when undergoing PHM treatment was aggravated again because the effect of the medications had worn out, rather than the phenomenon specifically occurring in the first pilot study because of the small sample size.

In the first pilot study, all three patients observed with sequestration–migration on MRI showed significant changes such that the MRI findings were no longer observed at the follow-up time point, whereas no significant changes were observed in the participants of this study. None of the patients underwent sequestration on MRI at screening, and among the eight patients (three in the non-PHM group and five in the PHM group) who showed disc extrusion, the MRI findings disappeared in two patients in the PHM group. In the main study, MRI scans will be performed over a longer follow-up period for more detailed monitoring, and an evaluation of LDH, etc., will be calculated.

Furthermore, in the first pilot study [27], three cases of AEs possibly related to the intervention, such as nausea and headaches, were observed in the PHM group; in the current pilot study, one case of AE, a headache, was observed in the PHM group. In contrast, no AEs with a possible relationship to the intervention were observed in the non-PHM group in either of the pilot studies. Non-steroidal anti-inflammatory drugs, such as aceclofenac, frequently used in this study, are associated with headaches and heartburn [33], and opioids, such as tramadol, are associated not only with problems such as abuse or misuse [34] but also with adverse drug reactions, including digestive symptoms, such as constipation and nausea, and neurologic symptoms, such as dizziness and headache [33]. Therefore, careful consideration of the benefits and disadvantages of each medication is necessary. On the other hand, the findings of our study reinforce the safety of spinal manipulation for patients with non-acute LDH. Despite concerns among some clinicians regarding the potential risks of spinal manipulation, previous research has also indicated that this treatment is generally safe for appropriately selected patients. A previous study showed that spinal manipulation did not negatively impact MRI-confirmed lumbar disc herniation [35]. Another cohort study found no significant association between spinal manipulation and cauda equina syndrome [36].

Analysis of the costs from a comparative economic evaluation between the two groups showed that the cost of intervention was higher in the non-PHM group than in the PHM group. Generally, the costs of medication and physiotherapy are low, whereas the costs of procedures and injections are high. As the percentage of patients undergoing procedures or injections was not high in the PHM group, the medical costs in the non-PHM group were calculated to be higher. The time–cost of intervention was also high in the non-PHM group for a similar reason that the percentage of the patients receiving procedures and injections, which take a longer treatment time than other types of treatment, was not high in the PHM group. During the follow-up period, there was no medical service utilization in the non-PHM group, whereas medical costs were incurred in the PHM group. This can be interpreted as meaning that the non-PHM group experienced good control of their symptoms, even after the end of the intervention period, thus requiring no additional medical services. The cost of productivity loss and the total cost, including the cost of productivity loss, were greater in the PHM group. In conclusion, from a societal perspective, the non-PHM treatment strategy was a dominant treatment strategy, which is more clinically and cost-effective than the PHM treatment strategies.

There are some limitations to the present study. The time to visit a pharmacy to receive the prescribed medications was not included in the patient survey in the process of calculating medical costs and time costs for economic evaluation, and the time–cost for additional medical service utilization other than the study interventions was not calculated. Since the study institutions were all KM hospitals, this may have had an impact on patient compliance and treatment effects in the PHM group. In fact, in the first pilot study performed at a single institution [27], some patients assigned to the PHM group expressed disappointment or were not cooperative in subsequent processes. Likewise, in this multicenter pilot study, participants expressed disappointment when they were assigned to the PHM group through the randomization process. Therefore, it was difficult to ensure that the participants in the PHM group continuously complied with study participation while receiving appropriate treatment in other medical institutions. Nevertheless, most of the participants faithfully attended treatment sessions or received necessary treatments according to the strategy to which they were assigned and brought their receipts recorded with details of the treatment they received without omission, confirming the feasibility of the main study with the PHM group.

In addition, the results of this pilot study should be interpreted with caution because power calculations were not performed in the analysis. Furthermore, while our findings provide valuable insights into the comparative effectiveness of non-PHM and PHM strategies in South Korea, the results may not be directly generalizable to other healthcare systems. The accessibility, utilization, and cost structures of non-PHM treatments, such as acupuncture and spinal manipulation, vary significantly across different countries. Therefore, it is not possible to conclude that non-PHM treatment strategies are superior to PHM treatment strategies based on the results of this study alone.

Exercise therapy is also one of the important non-pharmacological treatment strategies, but this study did not include exercise therapy because it mainly consisted of non-pharmacological treatments commonly used in Korean medicine. Future studies evaluating the effectiveness of non-pharmacological treatments, including exercise therapy, will also be necessary. To make the study more complete, it may also be considered to include a third control group, which has no treatment of any kind.

Despite these limitations, through this multicenter pilot study, the factors requiring consideration in performing the main study were identified, and information useful for designing the main study was acquired. In particular, the feasibility of patient recruitment and compliance was confirmed, and this study also served as an opportunity for researchers who were not familiar with pragmatic controlled trials, particularly the investigators of each study institution, to better understand and adapt to the study design.

In conclusion, we confirm the possibility that a non-PHM strategy could be a more effective and cost-effective treatment option than PHM for patients with non-acute lumbar disc herniation. The findings of this study will be useful for patients and clinicians as they consider many different options for optimal treatment and will be utilized as evidence and reference data for planning a large-scale main study in the future.

## Figures and Tables

**Figure 1 jcm-14-01204-f001:**
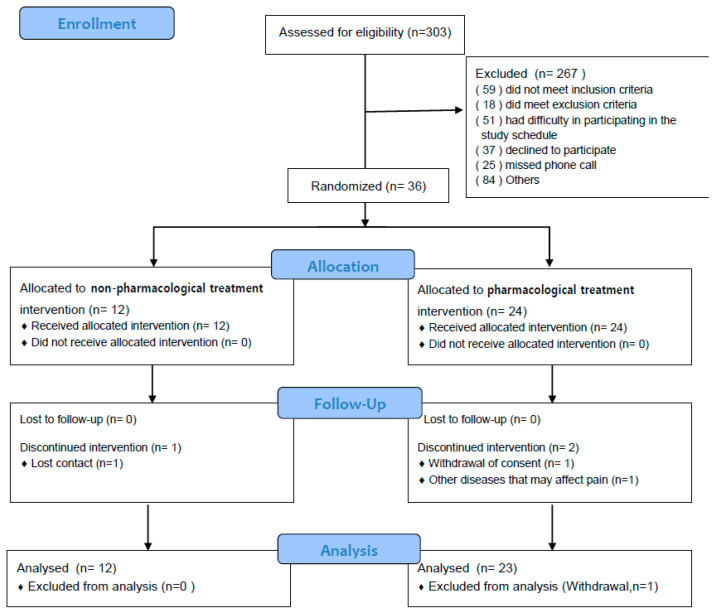
Flowchart of participants.

**Figure 2 jcm-14-01204-f002:**
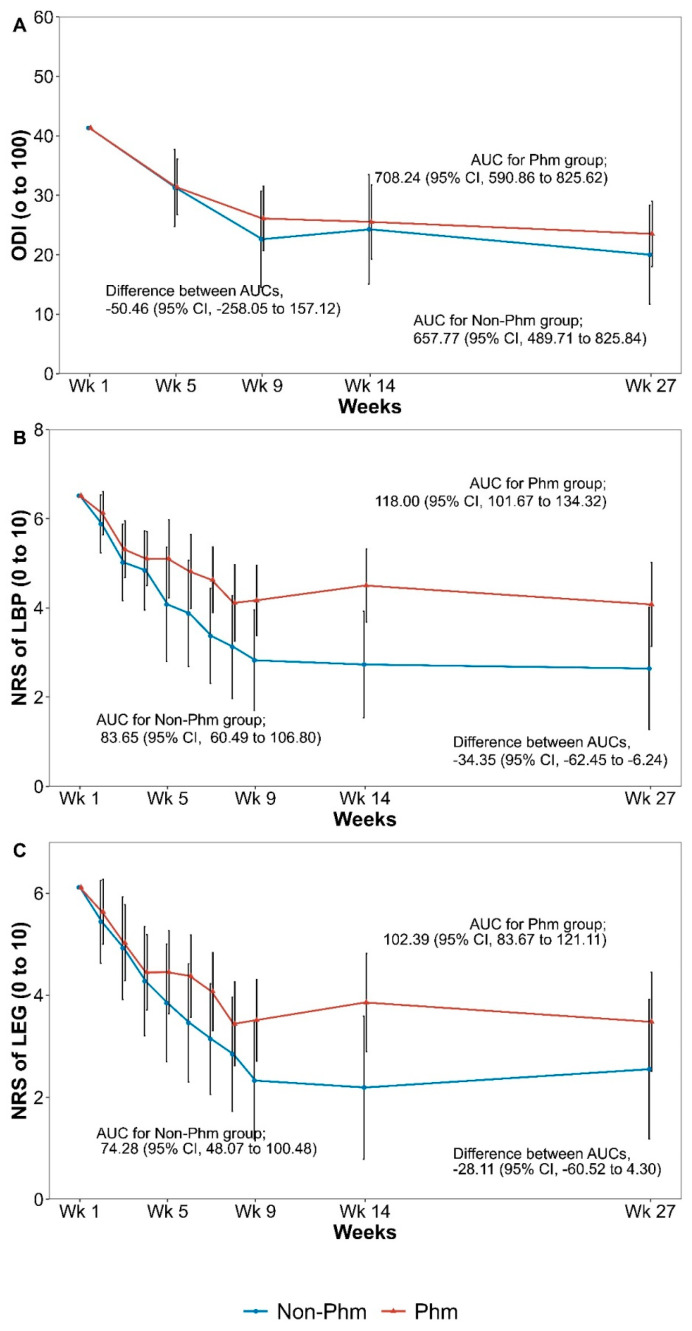
Area under the curves of each outcome by treatment group. (**A**) Oswestry disability index, (**B**) NRS of low back pain, (**C**) NRS for leg pain. AUC, area under the curve; PHM, pharmacological; Wk, week; ODI, Oswestry Disability Index; NRS, numeric rating scale; VAS, visual analog scale.

**Figure 3 jcm-14-01204-f003:**
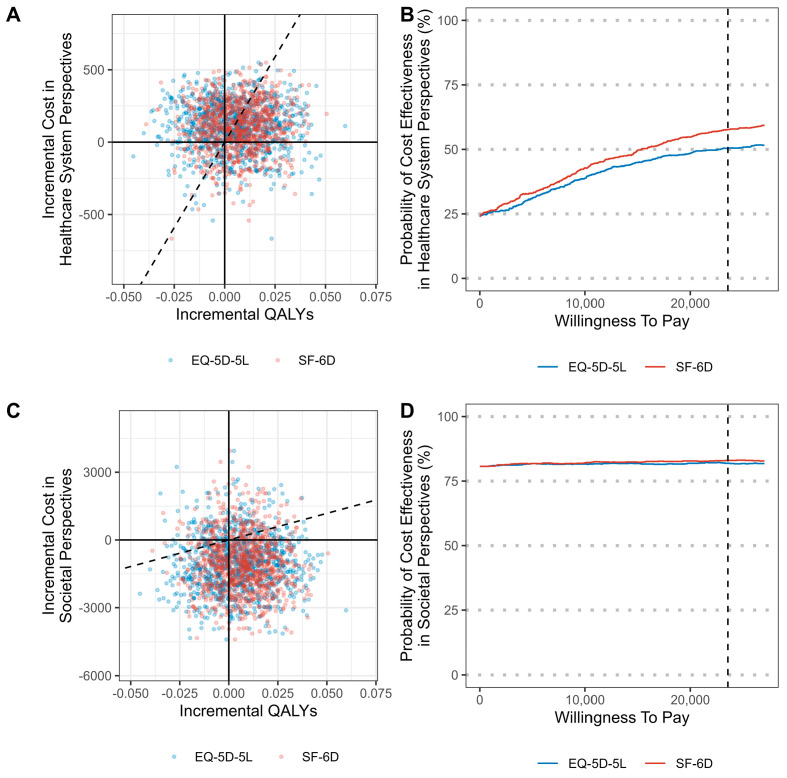
Cost-effectiveness plane and acceptability curves of non-pharmacological treatments compared with pharmacological treatments. Cost-effectiveness plane (**A**,**C**) and acceptability curves (**B**,**D**) comparing non-pharmacological and pharmacological treatment strategies. In the cost-effectiveness plane (**A**,**C**), each point represents a bootstrap sample of the incremental cost-effectiveness ratio. The *x*-axis indicates the difference in QALYs, and the *y*-axis represents the difference in cost. The diagonal dashed line represents the willingness-to-pay threshold. The number of points below the dashed line among the total number of points is the probability that non-PHM is cost-effective at that WTP. The acceptability curves (**B**,**D**) show the probability that the non-PHM treatment is cost-effective at different WTP values. EQ-5D-5L, EuroQoL 5-dimension 5-level instrument; SF-6D, Short-Form 6-Dimension; QALY, quality-adjusted life year.

**Table 1 jcm-14-01204-t001:** Baseline characteristics of participants by randomized group.

	Non-PHM (N = 12)	PHM (N = 23)	Total (N = 35)	*p*-Value
	N (%) or Mean (±SD)			
Sex				
Male (1)	6 (50.0%)	10 (43.5%)	16 (45.7%)	0.992
Female (2)	6 (50.0%)	13 (56.5%)	19 (54.3%)	
Age (years)	52.08 ± 10.98	44.26 ± 11.05	46.94 ± 11.50	0.055
Height (cm)	167.62 ± 9.42	166.57 ± 10.03	166.93 ± 9.70	0.767
Weight (kg)	64.57 ± 15.86	67.50 ± 14.70	66.49 ± 14.94	0.589
BMI	22.69 ± 3.53	24.09 ± 3.34	23.61 ± 3.42	0.254
Current employment status				
No	8 (66.7%)	8 (34.8%)	16 (45.7%)	0.150
Yes	4 (33.3%)	15 (65.2%)	19 (54.3%)	
Expectations for treatment scale				
Non-PHM	7.42 ± 1.16	6.70 ± 1.49	6.94 ± 1.41	0.155
PHM	6.00 ± 1.60	5.65 ± 1.23	5.77 ± 1.35	0.478
Treatment—steroid injections (times)				
0	11 (91.7%)	19 (82.6%)	30 (85.7%)	1.000
1	1 (8.3%)	2 (8.7%)	3 (8.6%)	
3	0 (0.0%)	1 (4.4%)	1 (2.9%)	
5	0 (0.0%)	1 (4.4%)	1 (2.9%)	
Treatment—procedures (times)				
0	11 (91.7%)	21 (91.3%)	32 (91.4%)	0.729
2	1 (8.3%)	0 (0.0%)	1 (2.9%)	
4	0 (0.0%)	1 (4.4%)	1 (2.9%)	
7	0 (0.0%)	1 (4.4%)	1 (2.9%)	
MRI				
Desiccated + bulging	12 (100.0%)	17 (73.9%)	29 (82.9%)	0.074
Protrusion	9 (75.0%)	19 (82.6%)	28 (80.0%)	0.67
Extrusion	4 (33.3%)	7 (30.4%)	11 (31.4%)	1.0
Sequestration + migration	0 (0.0%)	0 (0.0%)	0 (0.0)	-
Baseline outcome				
NRS—LBP	6.25 ± 1.60	6.65 ± 1.03	6.51 ± 1.25	0.372
NRS—leg pain	6.08 ± 0.90	6.13 ± 1.25	6.11 ± 1.13	0.909
VAS—LBP	60.92 ± 11.61	61.74 ± 12.94	61.46 ± 12.33	0.855
VAS—leg pain	62.92 ± 15.97	68.78 ± 8.90	66.77 ± 11.91	0.257
ODI	44.31 ± 10.95	39.74 ± 18.64	41.31 ± 16.38	0.441
EQ-5D-5L	0.62 ± 0.14	0.65 ± 0.18	0.64 ± 0.17	0.648
PCS	35.53 ± 5.02	38.10 ± 6.51	37.22 ± 6.10	0.241
MCS	46.14 ± 8.86	45.88 ± 10.68	45.97 ± 9.96	0.943

For treatment status, details of the treatment received within 3 months of screening were collected from patient surveys. Information on the types of procedures, such as laser therapy or minimally invasive procedures, excluding surgery and simple injection therapy, was collected. MRI was performed on all participants at screening and the last follow-up visit. The MRI images were read and interpreted by radiologists. PHM, pharmacological; BMI, body mass index; NRS, numeric rating scale; ODI, Oswestry Disability Index; VAS, visual analog scale; EQ-5D-5L, EuroQoL 5-dimension 5-level instrument; MCS, mental component summary; PCS, physical component summary.

**Table 2 jcm-14-01204-t002:** Treatment details in each group.

Group	Treatment Types	Total	Frequency per Patient (Mean ± SD)
Non-pharmacological treatment group (*n* = 12)	Acupuncture	12 (100)	10.8 ± 4.6
	Electroacupuncture	12 (100)	10.8 ± 4.6
	Chuna manual therapy	12 (100)	10.7 ± 4.6
	Transcutaneous infrared phototherapy	12 (100)	10.1 ± 4.6
	Cupping	7 (58.3)	12.9 ± 3.4
Pharmacological treatment group (*n* = 22)	Nerve blocks	9 (40.9)	3.4 ± 4.0
	Injections	6 (27.3)	2.5 ± 1.2
	Medications	18 (78.3)	26.6 ± 15.7
	Heat therapy	19 (86.4)	9.6 ± 6.8
	Electrotherapy	16 (72.7)	5.6. ± 3.4
	Manual therapy	14 (63.6)	6.6 ± 5.2
	Laser therapy	7 (31.8)	2.9 ± 2.5
	Extracorporeal shock wave therapy	3 (13.6)	3.7 ± 3.1
	Others	3 (13.6)	4.3 ± 2.5

The frequency of medications was calculated using the average prescription days of oral medications.

**Table 3 jcm-14-01204-t003:** Primary and secondary outcome changes according to the treatment group.

		Baseline	Week 5	Week 9 (Primary Endpoint)	Week 14	Week 27
ODI	Non-PHM	41.31 (35.88 to 46.74)	28.88 (21.50 to 36.27)	20.62 (13.24 to 28.00)	21.00 (13.62 to 28.39)	17.35 (9.97 to 24.73)
	PHM		31.67 (26.53 to 36.81)	25.79 (20.65 to 30.93)	25.44 (20.30 to 30.58)	23.91 (18.77 to 29.05)
	Difference	-	2.79 (−6.39 to 11.96)	5.17 (−4.00 to 14.35)	4.43 (−4.74 to 13.61)	6.57 (−2.61 to 15.74)
	*p*-value	-	0.544	0.262	0.336	0.157
NRS for LBP	Non-PHM	6.51(6.10 to 6.93)	3.72 (2.76 to 4.68)	2.45 (1.48 to 3.41)	2.36 (1.39 to 3.32)	2.36 (1.39 to 3.32)
	PHM		5.24 (4.56 to 5.92)	4.33 (3.65 to 5.01)	4.65 (3.97 to 5.33)	4.24 (3.56 to 4.92)
	Difference	-	1.52 (0.31 to 2.73)	1.89 (0.68 to 3.10)	2.30 (1.09 to 3.50)	1.89 (0.68 to 3.10)
	*p*-value	-	0.014	0.003	<0.001	0.003
NRS for leg	Non-PHM	6.11 (5.74 to 6.49)	3.48 (2.48 to 4.48)	2.03 (1.03 to 3.02)	1.75 (0.76 to 2.75)	2.21 (1.21 to 3.20)
	PHM		4.50 (3.78 to 5.21)	3.54 (2.83 to 4.25)	4.00 (3.28 to 4.71)	3.50 (2.78 to 4.21)
	Difference	-	1.02 (−0.23 to 2.27)	1.52 (0.27 to 2.77)	2.24 (0.99 to 3.49)	1.29 (0.04 to 2.54)
	*p*-value	-	0.109	0.018	<0.001	0.044
VAS for LBP	Non-PHM	61.46 (57.37 to 65.54)	32.80 (22.31 to 43.29)	17.34 (6.85 to 27.83)	16.26 (5.59 to 26.94)	20.06 (9.39 to 30.74)
	PHM		46.68 (39.20 to 54.15)	35.54 (28.07 to 43.01)	39.48 (31.88 to 47.09)	33.16 (25.40 to 40.92)
	Difference	-	13.88 (0.76 to 27.00)	18.20 (5.07 to 31.32)	23.22 (9.88 to 36.56)	13.10 (−0.33 to 26.52)
	*p*-value	-	0.039	0.007	<0.001	0.056
VAS for leg	Non-PHM	66.77 (62.83 to 70.72)	37.58 (27.68 to 47.48)	23.49 (13.59 to 33.39)	22.34 (12.21 to 32.47)	23.34 (13.21 to 33.47)
	PHM		53.22 (46.27 to 60.16)	44.49 (37.55 to 51.44)	46.09 (38.97 to 53.20)	41.75 (34.44 to 49.07)
	Difference	-	15.64 (3.20 to 28.07)	21.00 (8.56 to 33.43)	23.75 (11.04 to 36.45)	18.41 (5.59 to 31.23)
	*p*-value	-	0.014	0.001	<0.001	0.005
EQ-5D-5L	Non-PHM	0.64 (0.58 to 0.69)	0.75 (0.67 to 0.82)	0.79 (0.72 to 0.87)	0.82 (0.75 to 0.89)	0.80 (0.73 to 0.88)
	PHM		0.72 (0.67 to 0.77)	0.77 (0.72 to 0.82)	0.76 (0.71 to 0.81)	0.77 (0.72 to 0.82)
	Difference	-	−0.03 (−0.12 to 0.06)	−0.02 (−0.11 to 0.07)	−0.06 (−0.15 to 0.03)	−0.04 (−0.13 to 0.05)
	*p*-value	-	0.5	0.635	0.166	0.419
PCS	Non-PHM	37.22 (35.20 to 39.24)	79.35 (74.92 to 83.77)	82.81 (78.39 to 87.24)	82.40 (77.98 to 86.82)	84.09 (79.67 to 88.51)
	PHM		78.28 (75.20 to 81.36)	79.09 (76.01 to 82.17)	77.77 (74.70 to 80.85)	80.77 (77.69 to 83.85)
	Difference	-	−1.06 (−6.56 to 4.43)	−3.73 (−9.22 to 1.77)	−4.62 (−10.12 to 0.87)	−3.32 (−8.81 to 2.18)
	*p*-value	-	0.699	0.179	0.097	0.231
MCS	Non-PHM	45.97 (42.67 to 49.27)	94.59 (89.79 to 99.38)	96.57 (91.78 to 101.37)	97.15 (92.35 to 101.94)	97.39 (92.59 to 102.18)
	PHM		92.52 (89.18 to 95.87)	93.63 (90.29 to 96.98)	94.37 (91.03 to 97.72)	94.16 (90.82 to 97.51)
	Difference	-	−2.06 (−8.01 to 3.89)	−2.94 (−8.89 to 3.01)	−2.78 (−8.72 to 3.17)	−3.22 (−9.17 to 2.72)
	*p*-value	-	0.491	0.327	0.354	0.283
PGIC	Non-PHM	-	-	1.93 (1.36 to 2.50)	2.38 (1.82 to 2.95)	2.48 (1.91 to 3.04)
	PHM	-	-	2.81 (2.41 to 3.20)	3.03 (2.64 to 3.43)	2.90 (2.50 to 3.29)
	Difference	-	-	0.88 (0.17 to 1.58)	0.65 (−0.06 to 1.36)	0.42 (−0.28 to 1.13)
	*p*-value	-	-	0.016	0.07	0.234

Outcomes were assessed as decreases from baseline. A linear mixed-model approach was used to analyze the differences between the two groups. Baseline measurements of each outcome and age were used as covariates. ODI, Oswestry Disability Index; NRS, numeric rating scale; VAS, visual analog scale; EQ-5D-5L, EuroQoL 5-dimension 5-level instrument; MCS, mental component summary; PCS, physical component summary; PGIC, Patient Global Impression of Change; PHM, pharmacological.

**Table 4 jcm-14-01204-t004:** Comparisons of costs and QALYs between two groups.

	Non-PHM (*n* = 12)	PHM (*n* = 22)	Difference	*p*-Value
	Mean (95% CI)	Mean (95% CI)	Mean (95% CI)	
Medical costs				
Intervention period	1103 (931 to 1225)	804 (650 to 962)	299 (44 to 505)	0.024
Follow-up	0 (0 to 0)	191 (10 to 485)	−191 (−485 to −10)	0.002
Total	1103 (927 to 1230)	995 (732 to 1309)	108 (−235 to 412)	0.48
Non-medical costs				
Transportation	14 (6 to 23)	4 (1 to 8)	10 (1 to 19)	0.03
Time loss for intervention	444 (375 to 521)	283 (213 to 359)	161 (52 to 259)	0.006
Total	458 (389 to 536)	287 (221 to 360)	171 (74 to 272)	0.002
Productivity loss				
Intervention period	2181 (1676 to 2674)	2631 (2292 to 2998)	−450 (−1070 to 173)	0.118
Follow-up	3750 (2056 to 5883)	4570 (3702 to 5494)	−820 (−2802 to 1573)	0.448
Total	5931 (3949 to 8335)	7201 (6122 to 8307)	−1270 (−3647 to 1486)	0.314
Total cost (Limited societal perspectives + productivity loss)				
Intervention period	3742 (3273 to 4218)	3722 (3275 to 4146)	20 (−641 to 659)	0.969
Follow-up	3750 (2055 to 5870)	4761 (3967 to 5657)	−1011 (−2887 to 1236)	0.322
Total	7493 (5511 to 9842)	8483 (7420 to 9671)	−990 (−3338 to 1676)	0.388
QALYs				
EQ-5D-5L	0.381 (0.353 to 0.410)	0.376 (0.356 to 0.396)	0.005 (−0.031 to 0.042)	0.774
SF-6D	0.347 (0.324 to 0.371)	0.340 (0.324 to 0.356)	0.007 (−0.023 to 0.037)	0.629

EQ-5D-5L, EuroQoL 5-dimension 5-level instrument; SF-6D, Short-Form 6-Dimension; QALYs, quality-adjusted life years; PHM, pharmacological.

**Table 5 jcm-14-01204-t005:** Results of economic evaluation.

Perspectives	Difference in Cost	QALY Measures	Difference in QALYs	ICER (USD)	INB at 1 × WTP	*p*
Primary analysis						
Healthcare system	108 (−235 to 412)	EQ-5D-5L	0.005 (−0.031 to 0.042)	20,926	8 (−840 to 836)	50.4
		SF-6D	0.007 (−0.023 to 0.037)	15,260	54 (−615 to 736)	57.7
Societal perspectives	−990 (−3338 to 1676)	EQ-5D-5L	0.005 (−0.031 to 0.042)	Dominant	1162 (−1497 to 3549)	81.9
		SF-6D	0.007 (−0.023 to 0.037)	Dominant	1208 (−1535 to 3650)	83
Per-protocol analysis						
Healthcare system	18 (−446 to 344)	EQ-5D-5L	0.012 (−0.029 to 0.053)	3506	275 (−666 to 1242)	71.1
		SF-6D	0.012 (−0.020 to 0.043)	2556	257 (−554 to 1056)	73.3
Societal perspectives	−1867 (−4152 to 456)	EQ-5D-5L	0.012 (−0.029 to 0.053)	Dominant	2146 (−354 to 4641)	94
		SF-6D	0.012 (−0.020 to 0.043)	Dominant	2128 (−287 to 4536)	95.8

EQ-5D-5L, EuroQoL 5-dimension 5-level instrument; SF-6D, Short-Form 6-Dimension; QALYs, quality-adjusted life years; ICER, incremental cost-effectiveness ratio; INB, incremental net benefit; WTP, willingness to pay; *p*, probability of cost-effectiveness at one WTP.

## Data Availability

The data presented in this study are available upon request from the corresponding author. The data are not publicly available owing to privacy/ethical restrictions.

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
