# Peer review of "Comparative Effectiveness of Non-Pharmacological and Pharmacological Treatments for Non-Acute Lumbar Disc Herniation: A Multicenter, Pragmatic, Randomized Controlled, Parallel-Grouped Pilot Study"

_jcm, 2025, doi:10.3390/jcm14041204_

Round 1
Reviewer 1 Report
Comments and Suggestions for Authors
I was excited to see such a thorough pilot study which has several strengths including protocol registration and adherence, several outcome measures including clinical, cost, and safety metrics. The manuscript is well written, the study is justified, and the methods are well explained. The level of detail is high. I am glad to see such a high level of reporting which also follows CONSORT. I have no major comments. My comments are only to improve the level of detail and clarity of the manuscript.
Minor comments
1. Drug consumption – you mention collecting these data but could emphasize it was just for the pharmacologic group. It seems you might not have collected the drug consumption data for the non-PHM group (maybe they could not take medications? Or were not prescribed medications?). Also, maybe emphasize that these were prescription medications? I do not see any common over the counter medications listed in the supplemental table.
2. Discussion – you could mention a brief comment about how the study findings may not be directly generalizable to other countries or healthcare settings in which non-PHM may not be as routinely available or the cost of procedures may vary.
3. Discussion – you could include a brief comment that your study findings reinforce the safety of spinal manipulation for patients with lumbar disc herniation. It’s an important topic because in some areas, clinicians may be hesitant to use the therapy due to concerns about exacerbating the disc issue. While there is little large-scale data on the topic, spinal manipulation does not appear to trigger or worsen symptoms related to disc herniation. Perhaps cite another study or two? Example: Ehrler, Marco, et al. "Symptomatic, MRI confirmed, lumbar disc herniations: a comparison of outcomes depending on the type and anatomical axial location of the hernia in patients treated with high-velocity, low-amplitude spinal manipulation." Journal of manipulative and physiological therapeutics 39.3 (2016): 192-199. // Trager, R. J., Baumann, A. N., Perez, J. A., Dusek, J. A., Perfecto, R. P. T., & Goertz, C. M. (2024). Association between chiropractic spinal manipulation and cauda equina syndrome in adults with low back pain: Retrospective cohort study of US academic health centers. Plos one, 19(3), e0299159.
Comments for clarity
1. Abstract - The results in the abstract could have measures of variance like standard deviation or confidence intervals and/or measures of significance like p-value. Without either of these, it’s difficult to determine if the between-group differences are meaningful or statistically significant.
2. Abstract – you mention you had 26 weeks of follow-up but later on you say there was 27 weeks. Maybe this is a typo? – “follow-up assessments were performed at weeks 9, 13, and 26”
3. Abstract – the first few sentences could be a little more concise – example – omit “strategies” and in “main study” delete “main”
4. Abstract – consider changing one instance of “patients” to “adults” – e.g., “treatment strategies for patients with non-acute lumbar disc” – patients could be “adults”
5. Abstract – The acronym “EQ-5D” is not explained
6. Abstract and end of Discussion – some of the language (i.e., “confirmed”) is strong for a pilot study. You mention confirming the efficacy and cost-effectiveness, yet as a pilot study, I am not sure this phrasing is ideal. Maybe you could weaken the language a bit to say “suggest” or “points to” or “favors” or “indicates potential” etc.. That being said, I think it is fine to say “confirmed” with respect to the feasibility.
7. Keywords – consider adding “spinal manipulation” as a keyword
8. Introduction – can you please separate the last paragraph into two? This sentence – “Therefore, we aimed to compare the clinical efficacy…” could be a distinct/separate paragraph at the end of the introduction.
9. Introduction – some of the details at the end of the Introduction could be deleted, such as the “ratio of 1:2 for the non-PHM and PHM treatment strategy groups” – this is redundant with the Methods.
10. Methods – were patients in the non-PHM group allowed to receive pharmacologic care as well? Or was this restricted. Were they allowed to take over-the-counter analgesics or nonsteroidal anti-inflammatory medications?
11. Throughout – “efficacy” – I think your use of the term “efficacy” might not be ideal. Consider changing this to “effectiveness.” This is because your study is more pragmatic and gives the treating clinicians flexibility to decide which approaches to use, so it is less strict and more real-world. Therefore, effectiveness is a more appropriate term in this scenario.
12. “microscopic procedures” – I am not sure if this is a typo – maybe you mean “minimally invasive pain management procedures” like ablations?
13. Baseline table (Table 1) – The table is great yet could you please clarify if the “Treatment (Y/N)_steroid injections” indicates previous treatments, and if so, over what time span? E.g., “within past 3 months”. I am also not clear what “Treatment (Y/N)_procedures (times)” refers to – maybe the number of treatments patients had over the past 3 months? Or is this during the trial? (if so, it’s not necessarily a baseline characteristic). Consider maybe at the top left (or wherever appropriate) that you can indicate that the values are n (%) or mean (±SD). You might add something like “Variable n (%) or mean (±SD)” in that blank cell at the top left. Last, consider indenting each of the sub-items like Male and Female, under Sex, and the sub-items under Expectations.
14. Table 2 – consider clarifying that “Frequency per patient” is ±SD, to avoid confusion with CIs or standard error.
15. Table 3 – I am a little confused about the “Baseline” column. Why are some rows blank? Why does the value line up with non-PHM in some cases, yet in others it’s between non-PHM and PHM? Is the baseline column needed as baseline values are already shown in Table 1 for both groups. Second, I wonder why you used different asterisks like *, **, or *** for the p-values. This might not be necessary since you define significance as 0.05 in the methods, so the exact value may not matter as long as it’s <0.05. I also wonder if you can use another special symbol on the Week 9 column header to indicate that this is your primary endpoint.
16. Figure 3 – Can you give a little more information in the caption to describe what is being plotted? It says “acceptability curves of non-pharmacological treatments compared with pharmacological treatments” – does this represent the between-group differences? It seems the colors represent two different measurement items rather than the two groups. Maybe also give a sense of directionality -- for instance you could indicate whether a higher value on the y or x indicates a better cost effectiveness? What is the diagonal dashed line in A and C? Last – although a bit redundant, consider explaining the acronyms in the caption.
17. Section – “3.7 Economic evaluation” – can you clarify for some of these cost estimates, whether the values for cost apply to the entire group or whether they represent a mean per patient value? For example, USD $108, USD $961, etc.
18. Section – “3.8 Sample size calculation for the main study” – maybe you can make this clearer by emphasizing that the calculation is for a follow-up randomized controlled trial? I did not immediately understand this, wondering if it was a post-hoc calculation for the present study. It makes sense now, but I think some readers might be confused.
19. Discussion – The paragraph starting with “As these difficulties and areas of improvement were identified, it was deemed” is a little long. I think you already covered some of the information in terms of justifying your current study, and why your prior study had limitations, in the Introduction. Although it’s well-written, you could make this more concise.
Author Response
[Reviewer 1]
Dear reviewer 1,
We appreciate for your kind words for our paper. We also appreciate for the time and effort to review for this paper. We have striven to incorporate them in our revised version of the manuscript. Your suggestions and questions were extremely helpful in improving our paper. In the following sections, please find our responses to each of your questions. Thank you very much.
I was excited to see such a thorough pilot study which has several strengths including protocol registration and adherence, several outcome measures including clinical, cost, and safety metrics. The manuscript is well written, the study is justified, and the methods are well explained. The level of detail is high. I am glad to see such a high level of reporting which also follows CONSORT. I have no major comments. My comments are only to improve the level of detail and clarity of the manuscript.
Minor comments
- Drug consumption – you mention collecting these data but could emphasize it was just for the pharmacologic group. It seems you might not have collected the drug consumption data for the non-PHM group (maybe they could not take medications? Or were not prescribed medications?). Also, maybe emphasize that these were prescription medications? I do not see any common over the counter medications listed in the supplemental table.
- Response: Thank you for pointing this out. Drug consumption was investigated for all patients, all medications taken. That is, all prescribed medications as well as over-the-counter medications were investigated for both the PHM and non-PHM groups. The relevant content has been revised in the main text. (p5, 5th paragraph)
“2.6.3 Drug consumption
The types and doses of medications (prescribed for the present illness or rescue medications) taken during the study period were checked through a patient survey at each visit. All prescribed medications as well as over-the-counter medications were investigated for both the PHM and non-PHM groups. Other treatments, such as physiotherapy and injection therapy, apart from medications taken, were recorded using the frequency of the treatments received. Information on drug consumption was collected during each visit.”
- Discussion – you could mention a brief comment about how the study findings may not be directly generalizable to other countries or healthcare settings in which non-PHM may not be as routinely available or the cost of procedures may vary.
- Response: We agree that our results may not be fully generalizable to countries where non-PHM treatments are less accessible or where procedural costs differ. We added a statement in the Discussion to acknowledge this limitation. (p7, 4th paragraph)
“In addition, the results of this pilot study should be interpreted with caution because power calculations were not performed in the analysis. Furthermore, while our findings provide valuable insights into the comparative effectiveness of non-PHM and PHM strategies in South Korea, the results may not be directly generalizable to other healthcare systems. The accessibility, utilization, and cost sturctures of non-PHM treatments, such as acupuncture and spinal manipulation, vary significantly across different contries. Therefore, it is not possible to conclude that non-PHM treatment strategies are superior to PHM treatment strategies based on the results of this study alone”
- Discussion – you could include a brief comment that your study findings reinforce the safety of spinal manipulation for patients with lumbar disc herniation. It’s an important topic because in some areas, clinicians may be hesitant to use the therapy due to concerns about exacerbating the disc issue. While there is little large-scale data on the topic, spinal manipulation does not appear to trigger or worsen symptoms related to disc herniation. Perhaps cite another study or two? Example: Ehrler, Marco, et al. "Symptomatic, MRI confirmed, lumbar disc herniations: a comparison of outcomes depending on the type and anatomical axial location of the hernia in patients treated with high-velocity, low-amplitude spinal manipulation." Journal of manipulative and physiological therapeutics3 (2016): 192-199. // Trager, R. J., Baumann, A. N., Perez, J. A., Dusek, J. A., Perfecto, R. P. T., & Goertz, C. M. (2024). Association between chiropractic spinal manipulation and cauda equina syndrome in adults with low back pain: Retrospective cohort study of US academic health centers. Plos one, 19(3), e0299159. (line 512-518)
- Response: This is an important consideration. We added a discussion point about the safety of spinal manipulation and reference relevant studies, including those suggested by the reviewer, to support this aspect of our findings. (p6, last paragraph)
“... In other hands, the findings of our study reinforce the safety of spinal manipulation for patients with non-acute LDH. Despite concerns among some clinicians regarding the potential risks of spinal manipulation, previous research has also indicated that this treatment is generally safe for appropriately selected patients. A previous study showed that spinal manipulation did not negatively impact MRI-confirmed lumbar disc herniation[35]. Another cohort study found no significant association between spinal manuplation and cauda equina syndrome[36].”
Comments for clarity
- Abstract - The results in the abstract could have measures of variance like standard deviation or confidence intervals and/or measures of significance like p-value. Without either of these, it’s difficult to determine if the between-group differences are meaningful or statistically significant.
- Response: We revised the abstract to incorporate confidence intervals and p-values to ensure that the statistical significance of the findings is clear.
- Abstract – you mention you had 26 weeks of follow-up but later on you say there was 27 weeks. Maybe this is a typo? – “follow-up assessments were performed at weeks 9, 13, and 26”
- Response: It was unified into 27 weeks.
- Abstract – the first few sentences could be a little more concise – example – omit “strategies” and in “main study” delete “main”
- Response: I've revised that sentence to be more concise.
- Abstract – consider changing one instance of “patients” to “adults” – e.g., “treatment strategies for patients with non-acute lumbar disc” – patients could be “adults”
- Response: Thank you for your suggestion. However, I think that many papers use the expression 'patients with', and it seems a bit more common and natural than adult. Can I just use the expression “patient”? Thank you.
- Abstract – The acronym “EQ-5D” is not explained
- Response: We added the full name of “EQ-5D”.
- Abstract and end of Discussion – some of the language (i.e., “confirmed”) is strong for a pilot study. You mention confirming the efficacy and cost-effectiveness, yet as a pilot study, I am not sure this phrasing is ideal. Maybe you could weaken the language a bit to say “suggest” or “points to” or “favors” or “indicates potential” etc.. That being said, I think it is fine to say “confirmed” with respect to the feasibility.
- Response: Thank you for your suggestion. We have changed the expression.
“Background/Objectives: We aimed to compare non-pharmacological (non-PHM) and pharmacological (PHM) treatment for patients with non-acute lumbar disc herniation (LDH) and determine the feasibility for a large-scale study. Methods: This was a two-armed, parallel, multicenter, pragmatic controlled trial performed in South Korea. All patients underwent magnetic resonance imaging (MRI) scans both at the screening stage and the last follow-up. Patients with LDH findings on MRI were randomly assigned to non-PHM and PHM groups. Treatment was administered twice a week for a total of 8 weeks, and follow-up assessments were performed at weeks 9, 13, and 27 post-randomization. The primary outcome was the Oswestry Disability Index (ODI) score. A linear mixed model was used for primary analysis from intention-to-treat perspectives. The incremental cost-effectiveness ratio (ICER) was calculated for economic evaluation. Results: Thirty-six patients were enrolled, and 35 were included in the final analysis. At Week 9, the difference in ODI scores between the two groups was 5.17 (95% CI: -4.00 to 14.35, p = 0.262), and numeric rating scale scores for lower back and leg pains were 1.89 (95% CI: 0.68 to 3.10, p = 0.003) and 1.52 (95% CI: 0.27 to 2.77, p = 0.018), respectively, confirming greater improvement in the non-PHM group than in the PHM group. The non-PHM group showed lower costs and higher quality-adjusted life years than did the PHM group. The ICER calculated using the EuroQoL-5 Dimension (EQ-5D) was USD $20,926. Conclusions: We confirmed the possibility that a non-PHM strategy could be a more effective and cost-effective treatment option than PHM for patients with non-acute lumbar disc herniation. Furthermore, this pilot study confirmed the feasibility of the main study in terms of design and patient compliance.”
- Keywords – consider adding “spinal manipulation” as a keyword
- Response: We added the keyword.
- Introduction – can you please separate the last paragraph into two? This sentence – “Therefore, we aimed to compare the clinical efficacy…” could be a distinct/separate paragraph at the end of the introduction.
- Introduction – some of the details at the end of the Introduction could be deleted, such as the “ratio of 1:2 for the non-PHM and PHM treatment strategy groups” – this is redundant with the Methods.
- Response: We restructured the last paragraph for better readability and remove unnecessary details already covered in the Methods section. (p3, 3rd paragraph)
“Therefore, it was decided not to place restrictions on treatment institutions and to allow patients to freely receive PHM treatment at the institution of their choice. In addition, to facilitate the process of patient recruitment required for a large-scale trial, four hospitals participated as study institutions rather than as single centers as in the previous study. Since new study institutions were added and the research team was not familiar with the new study process of allowing treatment from external institutions other than the study institutions, the necessity for additional evaluation and confirmation of feasibility was suggested. Accordingly, we performed a comparative evaluation between non-PHM and PHM treatment strategies for patients with non-acute LDH with LBP or sciatica (radiating leg pain), and as a pilot study, aimed to evaluate the feasibility of conducting the subsequent main trial.”
- Methods – were patients in the non-PHM group allowed to receive pharmacologic care as well? Or was this restricted. Were they allowed to take over-the-counter analgesics or nonsteroidal anti-inflammatory medications?
- Response: Thank you for your good question. Because this study was a pragmatic clinical trial, patients in each group were recommended to receive the assigned treatment, but there were no specific restrictions on receiving additional treatments other than the recommended treatment strategy. This information was added to the Methods. (p4, 2nd paragraph)
“Patients assigned to each group were first educated about non-PHM and PHM treatment strategies and then received the applicable treatments. Next, according to the respective treatment strategies, patients received treatment based on the medical judgment of clinicians from the study and external medical institutions. Patients in each group were recommended to receive the assigned treatment, but there were no specific restrictions on receiving additional treatments other than the recommended treatment strategy.”
- Throughout – “efficacy” – I think your use of the term “efficacy” might not be ideal. Consider changing this to “effectiveness.” This is because your study is more pragmatic and gives the treating clinicians flexibility to decide which approaches to use, so it is less strict and more real-world. Therefore, effectiveness is a more appropriate term in this scenario.
- Response: Thank you for your opinion. We changed ‘efficacy’ into ‘effectiveness”.
- “microscopic procedures” – I am not sure if this is a typo – maybe you mean “minimally invasive pain management procedures” like ablations?
- Response: Yes, I mean “minimally invasive procedure”. I revised the word.
- Baseline table (Table 1) – The table is great yet could you please clarify if the “Treatment (Y/N)_steroid injections” indicates previous treatments, and if so, over what time span? E.g., “within past 3 months”. I am also not clear what “Treatment (Y/N)_procedures (times)” refers to – maybe the number of treatments patients had over the past 3 months? Or is this during the trial? (if so, it’s not necessarily a baseline characteristic). Consider maybe at the top left (or wherever appropriate) that you can indicate that the values are n (%) or mean (±SD). You might add something like “Variable n (%) or mean (±SD)” in that blank cell at the top left. Last, consider indenting each of the sub-items like Male and Female, under Sex, and the sub-items under Expectations
- Response: The footnote for the table was moved to the main text. I moved this back to a footnotes. The footnote states that the treatment history was investigated for 3 months prior to screening. In addition, as the reviewer suggested, "N (%) or mean (±SD)" was specified, and the sub-time was indented to increase the visibility of the table. (Table 1)
- Table 2 – consider clarifying that “Frequency per patient” is ±SD, to avoid confusion with CIs or standard error.
- Response: We revised table as you said.(Table2)
- Table 3 – I am a little confused about the “Baseline” column. Why are some rows blank? Why does the value line up with non-PHM in some cases, yet in others it’s between non-PHM and PHM? Is the baseline column needed as baseline values are already shown in Table 1 for both groups. Second, I wonder why you used different asterisks like *, **, or *** for the p-values. This might not be necessary since you define significance as 0.05 in the methods, so the exact value may not matter as long as it’s <0.05. I also wonder if you can use another special symbol on the Week 9 column header to indicate that this is your primary endpoint.
- Response: Since the baseline value was pooled from the two groups, the two cells were merged to show only one value. The asterisks were removed. It indicates that week 9 is the primary endpoint. (Table3)
- Figure 3 – Can you give a little more information in the caption to describe what is being plotted? It says “acceptability curves of non-pharmacological treatments compared with pharmacological treatments” – does this represent the between-group differences? It seems the colors represent two different measurement items rather than the two groups. Maybe also give a sense of directionality -- for instance you could indicate whether a higher value on the y or x indicates a better cost effectiveness? What is the diagonal dashed line in A and C? Last – although a bit redundant, consider explaining the acronyms in the caption.
- Response: We apologize for poor explanation. We added detail caption below the figure 3.
“Figure 3. Cost-effectiveness plane and acceptability curves of non-pharmacological treatments compared with pharmacological treatments. Cost-effectiveness plane (A, C) and acceptability curves (B, D) comparing non-pharmacological and pharmacological treatment strategies. In the cost-effectiveness plane (A, C), each point represents a bootstrap sample of the incremental cost-effectiveness ratio. The x-axis indicates the difference in QALYs, and the y-axis represents the difference in cost. The diagonal dashed line represents the willingness-to-pay threshold. The number of points below the dashed line among the total number of points is the probability that non-PHM is cost-effective at that WTP. The acceptability curves (B, D) show the probability that the non-PHM treatment is cost-effective at different WTP values. EQ-5D-5L, EuroQoL 5-Dimension 5-Level; SF-6D,Short Form-6 Dimension; QALY, Quality adusted life year.”
- Section – “3.7 Economic evaluation” – can you clarify for some of these cost estimates, whether the values for cost apply to the entire group or whether they represent a mean per patient value? For example, USD $108, USD $961, etc.
- Response: They represent a mean per patient. It is stated on the table that the values ​​are the mean and 95%CI.
- Section – “3.8 Sample size calculation for the main study” – maybe you can make this clearer by emphasizing that the calculation is for a follow-up randomized controlled trial? I did not immediately understand this, wondering if it was a post-hoc calculation for the present study. It makes sense now, but I think some readers might be confused.
- Response: Yes, this is the sample size calculation for the follow-up main study. As the reviewer said, it seems like it could be quite confusing. I deleted the content and will describe it in the protocol or results paper of the main study.
- Discussion – The paragraph starting with “As these difficulties and areas of improvement were identified, it was deemed” is a little long. I think you already covered some of the information in terms of justifying your current study, and why your prior study had limitations, in the Introduction. Although it’s well-written, you could make this more concise.
- Response: We deleted redundant sentences and reorganized the paragraph to make the paragraph more concise.
“As these difficulties and areas of improvement were identified, it was deemed necessary to conduct another pilot study. In particular, it was necessary to examine patient compliance, such as whether the patients in PHM group were properly receiving treatment at external medical institutions according to the treatment strategy to which they were assigned, and whether the treatment details at external medical institutions could be recorded and collected without problems. That is, since there was a greater need to verify feasibility in the PHM treatment (control) group than in the non-PHM treatment (experimental) group, the ratio of the non-PHM group to the PHM group was set at 1:2 for this multicenter pilot study. “

Reviewer 2 Report
Comments and Suggestions for Authors
Congratulations on the article; I found it to be interesting. The study's design is appropriate and engaging, well-written, with well-developed and visually appealing graphs. The statistical component is robust and well-supported; however, concerning the methodology, it may be advisable to include a third control group that does not receive treatment to observe the evolution.
I would like to offer some comments that I believe could enhance future studies:
· Firstly, I think the exclusive use of MRI should be reconsidered, and a neuropathic pain scale or neurodynamic/electrodiagnostic tests should be included.
· Regarding the treatment, it appears that some non-pharmacological interventions are mixed within the pharmacological group, which may bias the results.
· I also believe that therapeutic exercise should be incorporated into the non-pharmacological treatment, as many guidelines are beginning to reflect its importance.
· Finally, the discussion should include a reflection on why significant improvements were only observed in the NRS and VAS, and not in the ODI, for example.
These and other minor issues are documented in the attached PDF, which includes my notes.
Best regards,

Author Response
[Reviewer 2]
Dear reviewer 2,
We appreciate for your kind words for our paper. We also appreciate for the time and effort to review for this paper. We have striven to incorporate them in our revised version of the manuscript. Your suggestions and questions were extremely helpful in improving our paper. In the following sections, please find our responses to each of your questions. Since all comments in the text were included in the PDF, we only responded to the comments in the PDF. Thank you very much.
- This was a statistically non-significant result.
- Response: We added 95% CI and p value in result of Abstracts.
- A herniated lumbar disc does not necessarily cause sciatica or sciatic pain
- Response: “Lumbar disc herniation is one of the most common causes of low back pain and sciatica" was added. (p2, 1st paragraph)
" ..Lumbar disc herniation (LDH) is one of the most common cause of low back pain and sciatica[11]…"
- Many clinical guidelines now also recommend therapeutic physical exercise for back pain and sciatica.
- Response: Exercise therapy is also mentioned in many guidelines and is known to be an important non-pharmacological treatment. However, this study did not focus on exercise therapy, so it was not mentioned. We added a mention of exercise therapy. (p2, 2nd paragraph)
"..The American College of Physicians (ACP) recommends that clinicians and patients should initially select non-pharmacological (non-PHM) treatments such as superficial heat therapy, massage, acupuncture, exercise therapy or spinal manipulation for patients with acute or chronic LBP with or without sciatica.."
- Are these treatments considered pharmacological or non-pharmacological?
- Resoponse: Both epidural injections and lumbar facet joint blocks were considered drug treatments because they involve injecting drugs.
- To take a broader view, a non-pharmacological therapy should include therapeutic exercise.
- Response: Exercise therapy is also an important non-pharmacological treatment, but this study did not include exercise therapy because it aimed to examine the effectiveness of non-pharmacological treatments frequently used in Korean medicine. The fact that exercise therapy was not included was added to the Discussion. (p7, 5th paragraph)
" Exercise therapy is also one of the important non-pharmacological treatment strategies, but this study did not include exercise therapy because it mainly consisted of non-pharmacological treatments commonly used in Korean medicine. Future studies evaluating the effectiveness of non-pharmacological treatments, including exercise therapy, will also be necessary…"
- Was this sample size based on any reason?
- Response: The basis for the sample size of this study is explained in the reference. In other words, this study is a pilot study to evaluate the feasibility of a follow-up study, and 12 patients, which is considered to be the minimum required for a pilot study, were assigned to each group, and assuming a dropout rate of 20%, 15 patients were recruited per group, a total of 30 patients.
- The study would be more complete if there were a third control group to which no treatment of any kind is given. In this way we can check if improvements are obtained with respect to the control group
- Response: Thank you for your good suggestion. We will reflect this matter in the design of this study after sufficient discussion within the research team. It was briefly mentioned in the discussion section. (p7, 5th paragraph)
".. To make the study more complete, it may also be considered to include a third control group which has no treatment of any kind."
- "There are radiculopathies that do not have to be caused by a herniated disc, just as there are herniated discs that do not have to cause pain or radiculopathy. This inclusion criterion may exclude people with radiculopathy not appreciable on MRI or include people who have a herniation but it is not the cause of their pain.
- Response: Of course, there are various causes of radiculopathy. The subjects of this study were patients with herniated discs. We attempted to recruit patients with back pain or radiating pain due to herniated discs. Therefore, we did not unconditionally register patients who had imaging findings of discs, but included them in the study when they showed imaging findings of discs that could explain the symptoms.
- They are very different techniques to be able to draw conclusions, it would be more appropriate to compare a particular technique
- Response: This study is a practical clinical study, and it did not attempt to analyze the effectiveness of a specific technique, but rather to compare non-pharmacological treatment strategies with pharmacological treatment strategies. It seems that research on each technique should be conducted separately in the future.
- Were the subjects in this group monitored to ensure that they did not take anti-inflammatory drugs or opiates during treatment?
- There could be a bias in the comparison due to the fact that participants in this group received both pharmacological and non-pharmacological treatment.
- Response: This study is a pragmatic clinical study. Therefore, we educated each group on the appropriate treatment and recommended them to receive the treatment, but did not restrict other treatments. However, all treatments received were collected. This information was added to the methods section. (p4, 2nd paragraph)
"Patients assigned to each group were first educated about non-PHM and PHM treatment strategies and then received the applicable treatments. Next, according to the respective treatment strategies, patients received treatment based on the medical judgment of clinicians from the study and external medical institutions. Patients in each group were recommended to receive the assigned treatment, but there were no specific restrictions on receiving additional treatments other than the recommended treatment strategy."
- If there was a participant who was not contacted, shouldn't he/she have been excluded?
- Response: Since this study's main analysis was ITT analysis, all participants except those who withdrew consent were included in the analysis.
- If there were 24 and one withdrew his consent and another was excluded... wouldn't there be 22?
- Response: Except for one participant who withdrew consent. The description was misleading. It has been revised. (p7, 3rd paragraph)
“One participant in the non-PHM group could not be contacted during the intervention period, one participant in the PHM group withdrew consent, and one participant was discontinued the intervention because a disease that may affect the pain outcome in this study was identified during treatment.”
- Did pharmacological and non-pharmacological treatment come together?
- Response: As mentioned above, we educated each group on the treatment strategy and recommended that they receive the corresponding treatment, but did not restrict other treatments. However, we investigated all the treatment records, and table 2 shows the detailed treatment records investigated in this way.
- "It is relevant that patients only improved statistically in the NRS and VAS, no other questionnaire revealed statistical improvement. Perhaps this is a point that should be reflected on in the discussion.
- Response: Thank you for your suggestion. In relation to this, it is described in the discussion as follows: “In a previous single-center pilot study, the outcome measured in ODI scores showed some differences between the two groups, but the NRS scores were similar between the two groups. The difference in the results of the two clinical studies with similar designs may have been due to small sample sizes.’
- This index did not show significant improvements according to the tables shown
- Response: Thank you for your point out. It was our mistake. “ODI” was deleted.
- After the end of the section there should be a point of conclusion.
- Response: Thank you for your point out. We added the conclusion sentence. (p8, 2nd paragraph)
“In conclusion, we confirmed the possibility that a non-PHM strategy could be a more effective and cost-effective treatment option than PHM for patients with non-acute lumbar disc herniation. The findings of this study will be useful for patients and clinicians as they consider many different options for optimal treatment, and will be utilized as evidence and reference data for planning a large-scale main study in the future.”
